# Point-Voxel CNN for Efficient 3D Deep Learning

**Zhijian Liu**[*]
MIT

**Haotian Tang**[*]
Shanghai Jiao Tong University

**Yujun Lin**
MIT

**Song Han**
MIT

## Abstract

We present Point-Voxel CNN (PVCNN) for efficient, fast 3D deep learning. Previous work processes 3D data using either voxel-based or point-based NN models. However, both approaches are computationally inefficient. The computation cost and memory footprints of the voxel-based models grow *cubically* with the input resolution, making it memory-prohibitive to scale up the resolution. As for point-based networks, up to 80% of the time is wasted on structuring the *sparse* data which have rather poor memory locality, not on the actual feature extraction. In this paper, we propose PVCNN that represents the 3D input data in *points* to reduce the memory consumption, while performing the convolutions in *voxels* to reduce the irregular, sparse data access and improve the locality. Our PVCNN model is both memory and computation efficient. Evaluated on semantic and part segmentation datasets, it achieves a much higher accuracy than the voxel-based baseline with **10×** GPU memory reduction; it also outperforms the state-of-the-art point-based models with **7×** measured speedup on average. Remarkably, the narrower version of PVCNN achieves **2×** speedup over PointNet (an extremely efficient model) on part and scene segmentation benchmarks with much higher accuracy. We validate the general effectiveness of PVCNN on 3D object detection: by replacing the primitives in Frustrum PointNet with PVConv, it outperforms Frustrum PointNet++ by up to **2.4%** mAP with **1.8×** measured speedup and **1.4×** GPU memory reduction.

## 1 Introduction

3D deep learning has received increased attention thanks to its wide applications: *e.g.*, AR/VR and autonomous driving. These applications need to interact with people in real time and therefore require low latency. However, edge devices (such as mobile phones and VR headsets) are tightly constrained by hardware resources and battery. Therefore, it is important to design efficient and fast 3D deep learning models for real-time applications on the edge.

Collected by the LiDAR sensors, 3D data usually comes in the format of point clouds. Conventionally, researchers rasterize the point cloud into voxel grids and process them using 3D volumetric convolutions [4, 33]. With low resolutions, there will be information loss during voxelization: multiple points will be merged together if they lie in the same grid. Therefore, a high-resolution representation is needed to preserve the fine details in the input data. However, the computational cost and memory requirement both increase *cubically* with voxel resolution. Thus, it is infeasible to train a voxel-based model with high-resolution inputs: *e.g.*, 3D-UNet [51] requires more than 10 GB of GPU memory on 64×64×64 inputs with batch size of 16, and the large memory footprint makes it rather difficult to scale beyond this resolution.

Recently, another stream of models attempt to directly process the input point clouds [17, 23, 30, 32]. These point-based models require much lower GPU memory than voxel-based models thanks to the sparse representation. However, they neglect the fact that the *random memory access* is also very inefficient. As the points are scattered over the entire 3D space in an irregular manner, processing

---

[*] indicates equal contributions. The first two authors are listed in the alphabetical order.

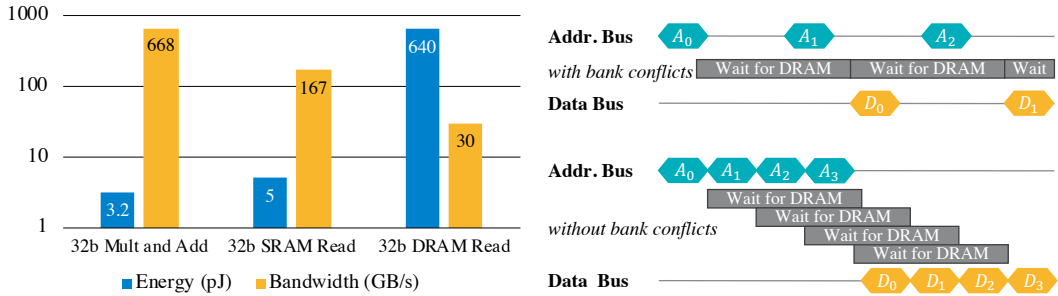

(a) Off-chip DRAM accesses take two orders of magnitude more energy than arithmetic operations (640pJ *vs*. 3pJ [10]), while the bandwidth is two orders of magnitude less (30GB/s *vs*. 668GB/s [16]). Efficient 3D deep learning should **reduce the memory footprint**, which is the bottleneck of conventional *voxel-based* methods.

(b) Random memory access is inefficient since it cannot take advantage of the DRAM burst and will cause bank conflicts [28], while contiguous memory access does not suffer from the above issue. Efficient 3D deep learning should **avoid random memory accesses**, which is the bottleneck of conventional *point-based* methods.

Figure 1: Efficient 3D models should reduce memory footprint and avoid random memory accesses.

them introduces random memory accesses. Most point-based models [23] mimic the 3D volumetric convolution: they extract the feature of each point by aggregating its neighboring features. However, neighbors are not stored contiguously in the point representation; therefore, indexing them requires the costly nearest neighbor search. To trade space for time, previous methods replicate the entire point cloud for each center point in the nearest neighbor search, and the memory cost will then be $\mathcal{O}(n^2)$, where $n$ is the number of input points. Another overhead is introduced by the dynamic kernel computation. Since the relative positions of neighbors are not fixed, these point-based models have to generate the convolution kernels dynamically based on different offsets.

Designing efficient 3D neural network models needs to take the hardware into account. Compared with arithmetic operations, memory operations are particularly expensive: they consume two orders of magnitude *higher* energy, having two orders of magnitude *lower* bandwidth (Figure 1a). Another aspect is the memory access pattern: the random access will introduce memory bank conflicts and decrease the throughput (Figure 1b). From the hardware perspective, conventional 3D models are inefficient due to large memory footprint and random memory access.

This paper provides a novel perspective to overcome these challenges. We propose Point-Voxel CNN (PVCNN) that represents the 3D input data as point clouds to take advantage of the sparsity to reduce the memory footprint, and leverages the voxel-based convolution to obtain the contiguous memory access pattern. Extensive experiments on multiple tasks demonstrate that PVCNN outperforms the voxel-based baseline with **10×** lower memory consumption. It also achieves **7×** measured speedup on average compared with the state-of-the-art point-based models.

## 2   Related Work

**Hardware-Efficient Deep Learning.**   Extensive attention has been paid to hardware-efficient deep learning for real-world applications. For instance, researchers have proposed to reduce the memory access cost by pruning and quantizing the models [7, 8, 9, 24, 39, 49] or directly designing the compact models [11, 12, 14, 25, 34, 48]. However, all these approaches are general-purpose and are suitable for arbitrary neural networks. In this paper, we instead design our efficient primitive based on some domain-specific properties: *e.g.*, 3D point clouds are highly sparse and spatially structured.

**Voxel-Based 3D Models.**   Conventionally, researchers relied on the volumetric representation to process the 3D data [45]. For instance, Maturana *et al*. [27] proposed the vanilla volumetric CNN; Qi *et al*. [31] extended 2D CNNs to 3D and systematically analyzed the relationship between 3D CNNs and multi-view CNNs; Wang *et al*. [40] incoporated the octree into volumetric CNNs to reduce the memory consumption. Recent studies suggest that the volumetric representation can also be used in 3D shape segmentation [21, 37, 44] and 3D object detection [50].

**Point-Based 3D Models.**   PointNet [30] takes advantage of the symmetric function to process the unordered point sets in 3D. Later research [17, 32, 43] proposed to stack PointNets hierarchically to model neighborhood information and increase model capacity. Instead of stacking PointNets as basic

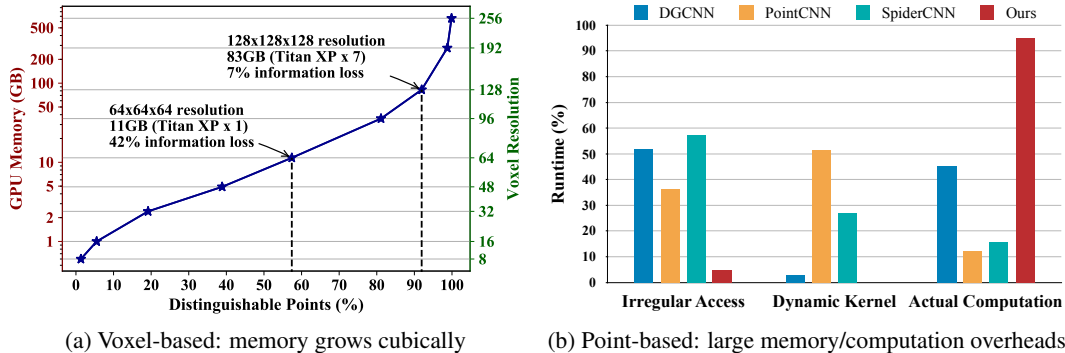

(a) Voxel-based: memory grows cubically

(b) Point-based: large memory/computation overheads

Figure 2: Both voxel-based and point-based NN models are inefficient. Left: the voxel-based model suffers from large information loss at acceptable GPU memory consumption (model: 3D-UNet [51]; dataset: ShapeNet Part [3]). Right: the point-based model suffers from large irregular memory access and dynamic kernel computation overheads.

blocks, another type of methods [18, 23, 46] abstract away the symmetric function using dynamically generated convolution kernels or learned neighborhood permutation function. Other research, such as SPLATNet [36] which naturally extends the idea of 2D image SPLAT to 3D, and SONet [22] which uses the self-organization mechanism with the theoretical guarantee of invariance to point order, also shows great potential in general-purpose 3D modeling with point clouds as input.

**Special-Purpose 3D Models.**   There are also 3D models tailored for specific tasks. For instance, SegCloud [38], SGPN [42], SPGraph [19], ParamConv [41], SSCN [6] and RSNet [13] are specialized in 3D semantic/instance segmentation. As for 3D object detection, F-PointNet [29] is based on the RGB detector and point-based regional proposal networks; PointRCNN [35] follows the similar idea while abstracting away the RGB detector; PointPillars [20] and SECOND [47] focus on the efficiency.

## 3   Motivation

3D data can be represented in the format of $\boldsymbol{x} = \{\boldsymbol{x}_k\} = \{(\boldsymbol{p}_k, \boldsymbol{f}_k)\}$, where $\boldsymbol{p}_k$ is the 3D coordinate of the $k^{\text{th}}$ input point or voxel grid, and $\boldsymbol{f}_k$ is the feature corresponding to $\boldsymbol{p}_k$. Both voxel-based and point-based convolution can then be formulated as

$$\boldsymbol{y}_k = \sum_{\boldsymbol{x}_i \in \mathcal{N}(\boldsymbol{x}_k)} \mathcal{K}(\boldsymbol{x}_k, \boldsymbol{x}_i) \times \mathcal{F}(\boldsymbol{x}_i). \tag{1}$$

During the convolution, we iterate the center $\boldsymbol{x}_k$ over the entire input. For each center, we first index its neighbors $\boldsymbol{x}_i$ in $\mathcal{N}(\boldsymbol{x}_k)$, then convolve the neighboring features $\mathcal{F}(\boldsymbol{x}_i)$ with the kernel $\mathcal{K}(\boldsymbol{x}_k, \boldsymbol{x}_i)$, and finally produces the corresponding output $\boldsymbol{y}_k$.

### 3.1   Voxel-Based Models: Large Memory Footprint

Voxel-based representation is regular and has good memory locality. However, it requires very high resolution in order not to lose information. When the resolution is low, multiple points are bucketed into the same voxel grid, and these points will no longer be *distinguishable*. A point is kept only when it exclusively occupies one voxel grid. In Figure 2a, we analyze the number of distinguishable points and the memory consumption (during training with batch size of 16) with different resolutions. On a single GPU (with 12 GB of memory), the largest affordable resolution is 64, which will lead to **42%** of information loss (*i.e.*, non-distinguishable points). To keep more than 90% of the information, we need to double the resolution to 128, consuming 7.2× GPU memory (**82.6 GB**), which is prohibitive for deployment. Although the GPU memory increases cubically with the resolution, the number of distinguishable points has a diminishing return. Therefore, the voxel-based solution is not scalable.

### 3.2   Point-Based Models: Irregular Memory Access and Dynamic Kernel Overhead

Point-based 3D modeling methods are memory efficient. The initial attempt, PointNet [30], is also computation efficient, but it lacks the local context modeling capability. Later research [23, 32, 43, 46]

improves the expressiveness of PointNet by aggregating the neighborhood information in the point domain. However, this will lead to the irregular memory access pattern and introduce the dynamic kernel computation overhead, which becomes the efficiency bottlenecks.

**Irregular Memory Access.**   Unlike the voxel-based representation, neighboring points $\boldsymbol{x}_i \in \mathcal{N}(\boldsymbol{x}_k)$ in the point-based representation are not laid out contiguously in memory. Besides, 3D points are scattered in $\mathbb{R}^3$; thus, we need to explicitly identify who are in the neighboring set $\mathcal{N}(\boldsymbol{x}_k)$, rather than by direct indexing. Point-based methods often define $\mathcal{N}(\boldsymbol{x}_k)$ as nearest neighbors in the coordinate space [23, 46] or feature space [43]. Either requires explicit and expensive KNN computation. After KNN, gathering all neighbors $\boldsymbol{x}_i$ in $\mathcal{N}(\boldsymbol{x}_k)$ requires large amount of random memory accesses, which is not cache friendly. Combining the cost of neighbor indexing and data movement, we summarize in Figure 2b that the point-based models spend **36**% [23], **52**% [43] and **57**% [46] of the total runtime on structuring the irregular data and random memory access.

**Dynamic Kernel Computation.**   For the 3D volumetric convolutions, the kernel $\mathcal{K}(\boldsymbol{x}_k, \boldsymbol{x}_i)$ can be directly indexed as the relative positions of the neighbor $\boldsymbol{x}_i$ are fixed for different center $\boldsymbol{x}_k$: *e.g.*, each axis of the coordinate offset $\boldsymbol{p}_i - \boldsymbol{p}_k$ can only be $0, \pm 1$ for the convolution with size of 3. However, for the point-based convolution, the points are scattered over the entire 3D space irregularly; therefore, the relative positions of neighbors become unpredictable, and we will have to calculate the kernel $\mathcal{K}(\boldsymbol{x}_k, \boldsymbol{x}_i)$ for each neighbor $\boldsymbol{x}_i$ *on the fly*. For instance, SpiderCNN [46] leverages the third-order Taylor expansion as a continuous approximation of the kernel $\mathcal{K}(\boldsymbol{x}_k, \boldsymbol{x}_i)$; PointCNN [23] permutes the neighboring points into a canonical order with the feature transformer $\mathcal{F}(\boldsymbol{x}_i)$. Both will introduce additional matrix multiplications. Empirically, we find that for PointCNN, the overhead of dynamic kernel computation can be more than **50**% (see Figure 2b)!

In summary, the combined overhead of irregular memory access and dynamic kernel computation ranges from **55**% (for DGCNN) to **88**% (for PointCNN), which indicates that most computations are wasted on dealing with the irregularity of the point-based representation.

## 4   Point-Voxel Convolution

Based on our analysis on the bottlenecks, we introduce a hardware-efficient primitive for 3D deep learning: Point-Voxel Convolution (PVConv), which combines the advantages of point-based methods (*i.e.*, small memory footprint) and voxel-based methods (*i.e.*, good data locality and regularity).

Our PVConv disentangles the *fine-grained* feature transformation and the *coarse-grained* neighbor aggregation so that each branch can be implemented efficiently and effectively. As illustrated in Figure 3, the upper voxel-based branch first transforms the points into *low-resolution* voxel grids, then it aggregates the neighboring points by the voxel-based convolutions, followed by devoxelization to convert them back to points. Either voxelization or devoxelization requires one scan over all points, making the memory cost low. The lower point-based branch extracts the features for each individual point. As it does not aggregate the neighbor's information, it is able to afford a very *high resolution*.

### 4.1   Voxel-Based Feature Aggregation

A key component of convolution is to aggregate the neighboring information to extract local features. We choose to perform this feature aggregation in the volumetric domain due to its regularity.

**Normalization.**   The scale of different point cloud might be significantly different. We therefore normalize the coordinates $\{\boldsymbol{p}_k\}$ before converting the point cloud into the volumetric domain. First, we translate all points into the local coordinate system with the gravity center as origin. After that, we normalize the points into the unit sphere by dividing all coordinates by $\max\|\boldsymbol{p}_k\|_2$, and we then scale and translate the points to $[0, 1]$. Note that the point features $\{\boldsymbol{f}_k\}$ remain unchanged during the normalization. We denote the normalized coordinates as $\{\hat{\boldsymbol{p}}_k\}$.

**Voxelization.**   We transform the normalized point cloud $\{(\hat{\boldsymbol{p}}_k, \boldsymbol{f}_k)\}$ into the voxel grids $\{\boldsymbol{V}_{u,v,w}\}$ by averaging all features $\boldsymbol{f}_k$ whose coordinate $\hat{\boldsymbol{p}}_k = (\hat{\boldsymbol{x}}_k, \hat{\boldsymbol{y}}_k, \hat{\boldsymbol{z}}_k)$ falls into the voxel grid $(u, v, w)$:

$$\boldsymbol{V}_{u,v,w,c} = \frac{1}{N_{u,v,w}} \sum_{k=1}^{n} \mathbb{I}[\text{floor}(\hat{\boldsymbol{x}}_k \times r) = u, \text{floor}(\hat{\boldsymbol{y}}_k \times r) = v, \text{floor}(\hat{\boldsymbol{z}}_k \times r) = w] \times \boldsymbol{f}_{k,c}, \quad (2)$$

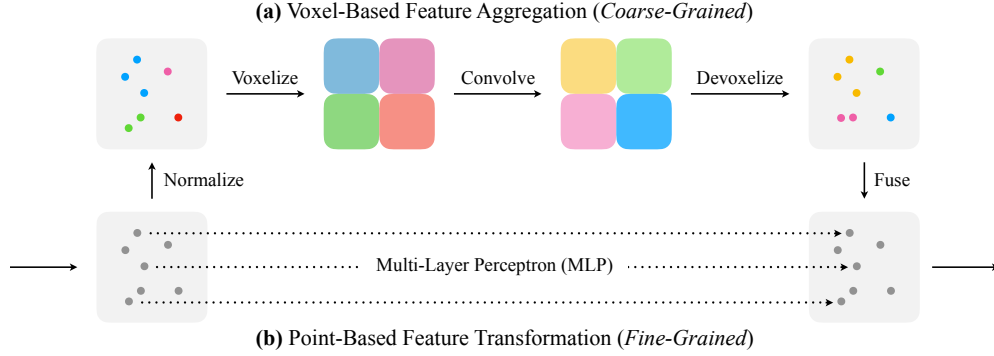

**(a)** Voxel-Based Feature Aggregation (*Coarse-Grained*)

**(b)** Point-Based Feature Transformation (*Fine-Grained*)

Figure 3: PVConv is composed of a *low-resolution* voxel-based branch and a *high-resolution* point-based branch. The voxel-based branch extracts *coarse-grained* neighborhood information, which is supplemented by the *fine-grained* individual point features extracted from the point-based branch.

where $r$ denotes the voxel resolution, $\mathbb{I}[\cdot]$ is the binary indicator of whether the coordinate $\hat{\boldsymbol{p}}_k$ belongs to the voxel grid $(u, v, w)$, $\boldsymbol{f}_{k,c}$ denotes the $c^{\text{th}}$ channel feature corresponding to $\hat{\boldsymbol{p}}_k$, and $N_{u,v,w}$ is the normalization factor (*i.e.*, the number of points that fall in that voxel grid). As the voxel resolution $r$ does not have to be large to be effective in our formulation (which will be justified in Section 5), the voxelized representation will not introduce very large memory footprint.

**Feature Aggregation.**    After converting the points into voxel grids, we apply a stack of 3D volumetric convolutions to aggregate the features. Similar to conventional 3D models, we apply the batch normalization [15] and the nonlinear activation function [26] after each 3D convolution.

**Devoxelization.**    As we need to fuse the information with the point-based feature transformation branch, we then transform the voxel-based features back to the domain of point cloud. A straightforward implementation of the voxel-to-point mapping is the nearest-neighbor interpolation (*i.e.*, assign the feature of a grid to all points that fall into the grid). However, this will make the points in the same voxel grid always share the same features. Therefore, we instead leverage the trilinear interpolation to transform the voxel grids to points to ensure that the features mapped to each point are distinct.

As our voxelization and devoxelization are both differentiable, the entire voxel-based feature aggregation branch can then be optimized in an end-to-end manner.

## 4.2    Point-Based Feature Transformation

The voxel-based feature aggregation branch fuses the neighborhood information in a coarse granularity. However, in order to model finer-grained individual point features, low-resolution voxel-based methods alone might not be enough. To this end, we directly operate on each point to extract individual point features using an MLP. Though simple, the MLP outputs distinct and discriminative features for each point. Such high-resolution individual point information is very critical to supplement the coarse-grained voxel-based information.

## 4.3    Feature Fusion

With both individual point features and aggregated neighborhood information, we can efficiently fuse two branches with an addition as they are providing complementary information.

## 4.4    Discussions

**Efficiency: Better Data Locality and Regularity.**    Our PVConv is more efficient than conventional point-based convolutions due to its better data locality and regularity. Our proposed voxelization and devoxelization both require $\mathcal{O}(n)$ random memory accesses, where $n$ is the number of points, since we only need to iterate over all points once to scatter them to their corresponding voxel grids. However, for conventional point-based methods, gathering the neighbors for all points requires at least $\mathcal{O}(kn)$ random memory accesses, where $k$ is the number of neighbors. Therefore, our PVCNN is $k\times$ more efficient from this viewpoint. As the typical value for $k$ is 32/64 in PointNet++ [32] and 16 in PointCNN [23], we empirically reduce the number of incontiguous memory accesses by $16\times$ to

|  | Input Data | Convolution | Mean IoU | Latency | GPU Memory |
|---|---|---|---|---|---|
| PointNet [30] | points (8×2048) | none | 83.7 | 21.7 ms | 1.5 GB |
| 3D-UNet [51] | voxels (8×96³) | volumetric | 84.6 | 682.1 ms | 8.8 GB |
| RSNet [13] | points (8×2048) | point-based | 84.9 | 74.6 ms | 0.8 GB |
| PointNet++ [32] | points (8×2048) | point-based | 85.1 | 77.9 ms | 2.0 GB |
| DGCNN [43] | points (8×2048) | point-based | 85.1 | 87.8 ms | 2.4 GB |
| **PVCNN** (Ours, 0.25×C) | points (8×2048) | volumetric | **85.2** | **11.6 ms** | **0.8 GB** |
| SpiderCNN [46] | points (8×2048) | point-based | 85.3 | 170.7 ms | 6.5 GB |
| **PVCNN** (Ours, 0.5×C) | points (8×2048) | volumetric | **85.5** | **21.7 ms** | **1.0 GB** |
| PointCNN [23] | points (8×2048) | point-based | 86.1 | 135.8 ms | 2.5 GB |
| **PVCNN** (Ours, 1×C) | points (8×2048) | volumetric | **86.2** | **50.7 ms** | **1.6 GB** |

Table 1: Results of object part segmentation on ShapeNet Part. On average, PVCNN outperforms the point-based models with **5.5×** measured speedup and **3×** memory reduction, and outperforms the voxel-based baseline with **59×** measured speedup and **11×** memory reduction.

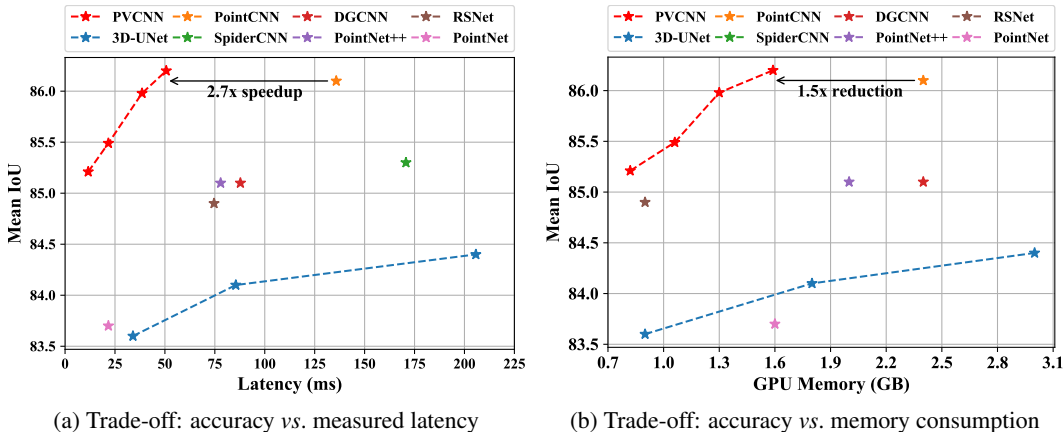

(a) Trade-off: accuracy *vs.* measured latency    (b) Trade-off: accuracy *vs.* memory consumption

Figure 4: Comparisons between PVCNN and point/voxel-based baselines on ShapeNet Part.

64× through our design and achieve better data locality. Besides, as our convolutions are done in the voxel domain, which is regular, our PVConv does not require KNN computation and dynamic kernel computation, which are usually quite expensive.

**Effectiveness: Keeping Points in High Resolution.**    As our point-based feature extraction branch is implemented as MLP, a natural advantage is that we are able to maintain the same number of points throughout the whole network while still having the capability to model neighborhood information. Let us make a comparison between our PVConv and set abstraction (SA) module in PointNet++ [32]. Suppose we have a batch of 2048 points with 64-channel features (with batch size of 16). We consider to aggregate information from 125 neighbors of each point and transform the aggregated feature to output the features with the same size. The SA module will require 75.2 ms of latency and 3.6 GB of memory consumption, while our PVConv will only require 25.7 ms of latency and 1.0 GB of memory consumption. The SA module will have to downsample to 685 points (*i.e.*, around 3× downsampling) to match up with the latency of our PVConv, while the memory consumption will still be 1.5× higher. Thus, with the same latency, our PVConv is capable of modeling the full point cloud, while the SA module has to downsample the input aggressively, which will inevitably induce information loss. Therefore, our PVCNN is more effective compared to its point-based counterpart.

## 5   Experiments

We experimented on multiple 3D tasks including object part segmentation, indoor scene segmentation and 3D object detection. Our PVCNN achieves superior performance on all these tasks with lower measured latency and GPU memory consumption. More details are provided in the appendix.

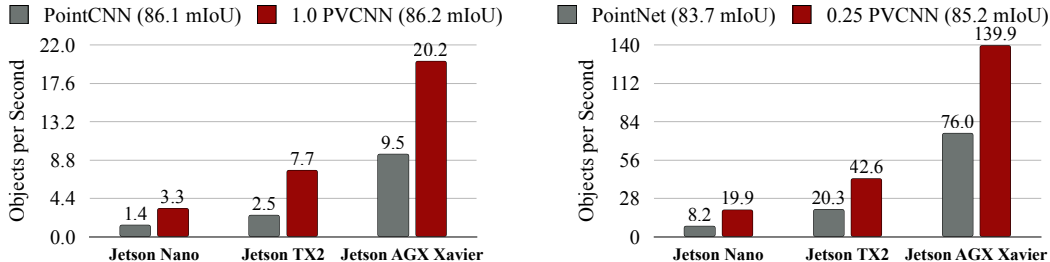

Figure 5: PVCNN runs efficiently on edge devices with low latency.

|  | mIoU | Latency | GPU Mem. |
|---|---|---|---|
| **PVCNN** (1×R) | 86.2 | 50.7 ms | 1.59 GB |
| **PVCNN** (0.75×R) | 85.7 | 36.8 ms | 1.56 GB |
| **PVCNN** (0.5×R) | 85.5 | 28.9 ms | 1.55 GB |

|  | ΔmIoU |
|---|---|
| Devoxelization w/o trilinear interpolation | -0.5 |
| 1× voxel convolution in each PVConv | -0.6 |
| 3× voxel convolution in each PVConv | -0.1 |

Table 2: Results of different voxel resolutions.    Table 3: Results of more ablation studies.

(a) Top row: features extracted from *coarse-grained* voxel-based branch (large, continuous).

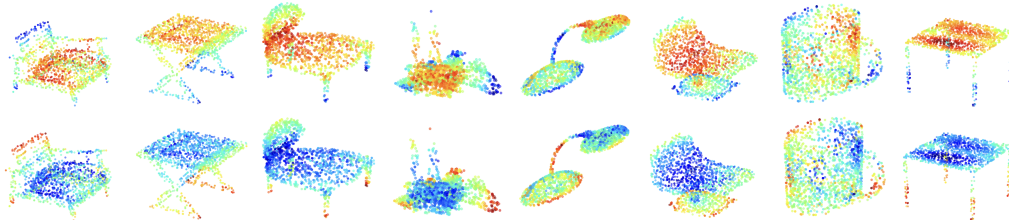

(b) Bottom row: features extracted from *fine-grained* point-based branch (isolated, discontinuous).

Figure 6: Two branches are providing complementary information: the voxel-based branch focuses on the large, continuous parts, while the point-based focuses on the isolated, discontinuous parts.

## 5.1 Object Part Segmentation

**Setups.** We first conduct experiments on the large-scale 3D object dataset, ShapeNet Parts [3]. For a fair comparison, we follow the same evaluation protocol as in Li *et al*. [23] and Graham *et al*. [6]. The evaluation metric is mean intersection-over-union (mIoU): we first calculate the part-averaged IoU for each of the 2874 test models and average the values as the final metrics. Besides, we report the measured latency and GPU memory consumption on a single GTX 1080Ti GPU to reflect the efficiency. We ensure the input data to have the same size with 2048 points and batch size of 8.

**Models.** We build our PVCNN by replacing the MLP layers in PointNet [30] with our PVConv layers. We adopt PointNet [30], RSNet [13], PointNet++ [32] (with multi-scale grouping), DGCNN [43], SpiderCNN [46] and PointCNN [23] as our point-based baselines. We reimplement 3D-UNet [51] as our voxel-based baseline. Note that most baselines make their implementation publicly available, and we therefore collect the statistics from their official implementation.

**Results.** As in Table 1, our PVCNN outperforms all previous models. PVCNN directly improves the accuracy of its backbone (PointNet) by 2.5% with even smaller overhead compared with PointNet++. We also design narrower versions of PVCNN by reducing the number of channels to 25% (denoted as 0.25×C) and 50% (denoted as 0.5×C). The resulting model requires only 53.5% latency of PointNet, and it still outperforms several point-based methods with sophisticated neighborhood aggregation including RSNet, PointNet++ and DGCNN, which are almost an order of magnitude slower.

In Figure 4, PVCNN achieves a significantly better accuracy *vs*. latency trade-off compared with all point-based methods. With similar accuracy, our PVCNN is **15×** faster than SpiderCNN and **2.7×** faster than PointCNN. Our PVCNN also achieves a significantly better accuracy *vs*. memory trade-off compared with modern voxel-based baseline. With better accuracy, PVCNN saves the GPU memory consumption by **10×** compared with 3D-UNet.

| | Input Data | Convolution | mAcc | mIoU | Latency | GPU Mem. |
|---|---|---|---|---|---|---|
| PointNet [30] | points (8×4096) | none | 82.54 | 42.97 | 20.9 ms | 1.0 GB |
| **PVCNN** (Ours, 0.125×C) | points (8×4096) | volumetric | **82.60** | **46.94** | **8.5 ms** | **0.6 GB** |
| DGCNN [43] | points (8×4096) | point-based | 83.64 | 47.94 | 178.1 ms | 2.4 GB |
| RSNet [13] | points (8×4096) | point-based | – | 51.93 | 111.5 ms | 1.1 GB |
| **PVCNN** (Ours, 0.25×C) | points (8×4096) | volumetric | **85.25** | **52.25** | **11.9 ms** | **0.7 GB** |
| 3D-UNet [51] | voxels (8×96³) | volumetric | 86.12 | 54.93 | 574.7 ms | 6.8 GB |
| **PVCNN** (Ours, 1×C) | points (8×4096) | volumetric | 86.66 | 56.12 | 47.3 ms | 1.3 GB |
| **PVCNN++** (Ours, 0.5×C) | points (4×8192) | volumetric | **86.87** | **57.63** | **41.1 ms** | **0.7 GB** |
| PointCNN [23] | points (16×2048) | point-based | 85.91 | 57.26 | 282.3 ms | 4.6 GB |
| **PVCNN++** (Ours, 1×C) | points (4×8192) | volumetric | **87.12** | **58.98** | **69.5 ms** | **0.8 GB** |

Table 4: Results of indoor scene segmentation on S3DIS. On average, our PVCNN and PVCNN++ outperform the point-based models with **8×** measured speedup and **3×** memory reduction, and outperform the voxel-based baseline with **14×** measured speedup and **10×** memory reduction.

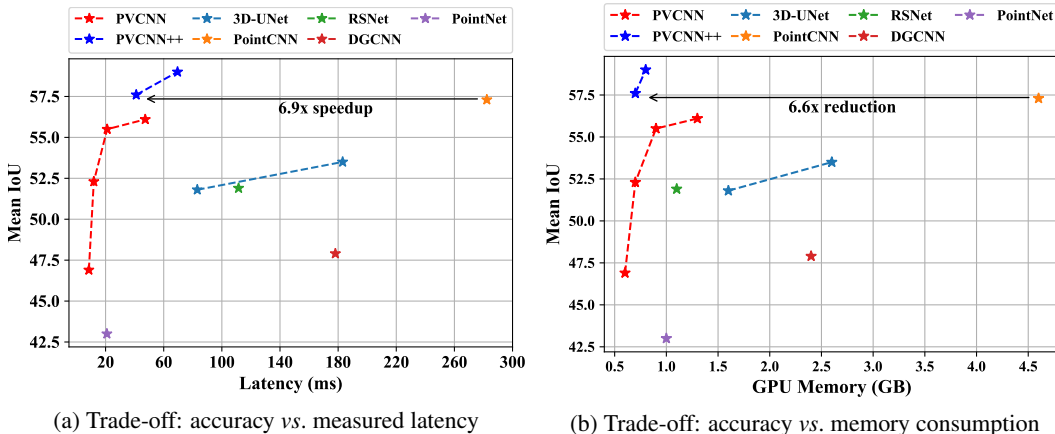

(a) Trade-off: accuracy *vs.* measured latency      (b) Trade-off: accuracy *vs.* memory consumption

Figure 7: Comparisons between PVCNN and point/voxel-based baselines on S3DIS.

Furthermore, we also measure the latency of PVCNN on three edge devices. In Figure 5, PVCNN consistently achieves a speedup of **2×** over PointNet and PointCNN on different devices. Especially, PVCNN is able to run at 19.9 objects per second on Jetson Nano with PointNet++-level accuracy and 20.2 objects per second on Jetson Xavier with PointCNN-level accuracy.

**Analysis.** Conventional voxel-based methods have saturated the performance as the input resolution increases, but the memory consumption grows cubically. PVCNN is much more efficient, and the memory increases sub-linearly (Table 2). By increasing the resolution from 16 (0.5×R) to 32 (1×R), the GPU memory usage is increased from 1.55 GB to 1.59 GB, only 1.03×. Even if we squeeze the volumetric resolution to 16 (0.5×R), our method still outperforms 3D-UNet that has much higher voxel resolution (96) by a large margin (1%). PVCNN is very robust even with small resolution in the voxel branch, thanks to the high-resolution point-based branch maintaining the individual point's information. We also compared different implementations of devoxelization in Table 3. The trilinear interpolation performs better than the nearest neighbor, which is because the points near the voxel boundaries will introduce larger fluctuations to the gradient, making it harder to optimize.

**Visualization.** We illustrate the voxel and point branch features from the final PVConv in Figure 6, where warmer color represents larger magnitude. We can see that the voxel branch captures large, continuous parts (*e.g.* table top, lamp head) while the point branch captures isolated, discontinuous details (*e.g.*, table legs, lamp neck). The two branches provide complementary information and can be explained by the fact that the convolution operation extracts features with continuity and locality.

## 5.2 Indoor Scene Segmentation

**Setups.** We conduct experiments on the large-scale indoor scene segmentation dataset, S3DIS [1,2]. We follow Tchapmi *et al.* [38] and Li *et al.* [23] to train the models on area 1,2,3,4,6 and test them on

| | Efficiency | | Car | | | Pedestrian | | | Cyclist | | |
|---|---|---|---|---|---|---|---|---|---|---|---|
| | Latency | GPU Mem. | Easy | Mod. | Hard | Easy | Mod. | Hard | Easy | Mod. | Hard |
| F-PointNet [29] | 29.1 ms | 1.3 GB | 85.24 | 71.63 | 63.79 | 66.44 | 56.90 | 50.43 | 77.14 | 56.46 | 52.79 |
| F-PointNet++ [29] | 105.2 ms | 2.0 GB | 84.72 | 71.99 | 64.20 | 68.40 | 60.03 | 52.61 | 75.56 | 56.74 | 53.33 |
| **F-PVCNN** (Ours) | 58.9 ms | 1.4 GB | **85.25** | **72.12** | **64.24** | **70.60** | **61.24** | **56.25** | **78.10** | **57.45** | **53.65** |

Table 5: Results of 3D object detection on the *val* set of KITTI. F-PVCNN outperforms F-PointNet++ in all categories significantly with **1.8×** measured speedup and **1.4×** memory reduction.

area 5 since it is the only area that does not overlap with any other area. Both data processing and evaluation protocol are the same as PointCNN [23] for fair comparison. We measure the latency and memory consumption with 32768 points per batch at test time on a single GTX 1080Ti GPU.

**Models.** Apart from PVCNN (which is based on PointNet), we also extend PointNet++ [32] with our PVConv to build PVCNN++. We compare our two models with the state-of-the-art point-based models [13, 23, 30, 43] and the voxel-based baseline [51].

**Results.** As in Table 4, PVCNN improves its backbone (PointNet) by more than **13%** in mIoU, and it also outperforms DGCNN (which involves sophisticated graph convolutions) by a large margin in both accuracy and latency. Remarkably, our PVCNN++ outperforms the state-of-the-art point-based model (PointCNN) by 1.7% in mIoU with **4×** lower latency, and the voxel-based baseline (3D-UNet) by 4% in mIoU with more than **8×** lower latency and GPU memory consumption.

Similar to object part segmentation, we design compact models by reducing the number of channels in PVCNN to 12.5%, 25% and 50% and PVCNN++ to 50%. Remarkably, the narrower version of our PVCNN outperforms DGCNN with **15×** measured speedup, and RSNet with **9×** measured speedup. Furthermore, it achieves 4% improvement in mIoU upon PointNet while still being **2.5×** faster than this extremely efficient model (which does not have any neighborhood aggregation).

### 5.3 3D Object Detection

**Setups.** We finally conduct experiments on the driving-oriented dataset, KITTI [5]. We follow Qi *et al*. [29] to construct the *val* set from the training set so that no instances in the *val* set belong to the same video clip of any training instance. The size of *val* set is 3769, leaving the other 3711 samples for training. We evaluate all models for 20 times and report the mean 3D average precision (AP).

**Models.** We build our F-PVCNN based on F-PointNet [29] by replacing the MLP layers within the instance segmentation network with our PVConv primitive and keep the box proposal and refinement networks unchanged. We compare our model with F-PointNet (whose backbone is PointNet) and F-PointNet++ (whose backbone is PointNet++). We report their results based on our reproduction.

**Results.** In Table 5, even if our F-PVCNN model does not aggregate neighboring features in the box estimation network while F-PointNet++ does, ours still outperforms it in all classes with **1.8×** lower latency. Specifically, our model achieves **2.4%** average mAP improvement in the most challenging pedestrian class. Compared with F-PointNet, our F-PVCNN obtains up to **4-5%** mAP improvement in pedestrians, which indicates that our proposed model is both efficient and expressive.

## 6 Conclusion

We propose Point-Voxel CNN (PVCNN) for fast and efficient 3D deep learning. We bring the best of both worlds together: voxels and points, reducing the memory footprint and irregular memory access. We represent the 3D input data efficiently with the sparse, irregular point representation and perform the convolutions efficiently in the dense, regular voxel representation. Extensive experiments on multiple tasks consistently demonstrate the effectiveness and efficiency of our proposed method. We believe that our research will break the stereotype that the voxel-based convolution is naturally inefficient and shed light on co-designing the voxel-based and point-based network architectures.

**Acknowledgements.** We thank MIT Quest for Intelligence, MIT-IBM Watson AI Lab, Samsung, Facebook and SONY for supporting this research. We also thank AWS Machine Learning Research Awards for providing the computation resource and NVIDIA for donating the Jetson AGX Xavier.

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
