[Supplementary Material]

## A.1 Model Details

Our PVCNN is mainly composed of the following two building primitives:

- `SharedLinear(in_channels, out_channels, activation)`:

  This module applies a *shared* linear transformation over all of the input points. It projects the features from `in_channels` to `out_channels` dimensions. The argument `activation` is a binary indicator of whether the batch normalization and non-linear activation are applied.

- `PVConv(in_channels, out_channels, kernel_size, voxel_size)`:

  This module is our proposed Point-Voxel Convolution (PVConv). For the voxel branch, we voxelize the point cloud with resolution of `voxel_size`, and we apply two 3D volumetric convolutions both with size of `kernel_size`.

We present the detailed PVCNN architectures for different 3D tasks in this section.

### A.1.1 PVCNN for Object Part Segmentation

For object part segmentation on ShapeNet Part [3], our PVCNN consumes 2048 points as input at each time. The model follows an encoder-decoder architecture. Specifically, the encoder consists of

```
PVConv(in_channels=6, out_channels=64, kernel_size=3, voxel_size=32)
PVConv(in_channels=64, out_channels=128, kernel_size=3, voxel_size=16)
PVConv(in_channels=128, out_channels=128, kernel_size=3, voxel_size=16)
SharedLinear(in_channels=128, out_channels=512)
SharedLinear(in_channels=512, out_channels=2048)
```

We then concatenate the outputs from all these layers together with the max-pooled 2048-dimension feature over all points and the 16-dimension one-hot object class label. The extracted feature has, in total, 4944 dimensions. Taking these features as input, the decoder is composed of

```
SharedLinear(in_channels=4944, out_channels=256)
SharedLinear(in_channels=256, out_channels=128)
SharedLinear(in_channels=128, out_channels=50, activation=False)
```

### A.1.2 PVCNN for Indoor Scene Segmentation

For indoor scene segmentation on S3DIS [1, 2], our PVCNN consumes 4096 points as input at each time. Similarly, the model follows an encoder-decoder architecture. The encoder is composed of

```
PVConv(in_channels=9, out_channels=64, kernel_size=3, voxel_size=32)
PVConv(in_channels=64, out_channels=64, kernel_size=3, voxel_size=16)
PVConv(in_channels=64, out_channels=128, kernel_size=3, voxel_size=16)
PVConv(in_channels=128, out_channels=128, kernel_size=3, voxel_size=16)
SharedLinear(in_channels=128, out_channels=1024)
```

We pass the 1024-dimension feature from the last layer through

```
SharedLinear(in_channels=1024, out_channels=256)
SharedLinear(in_channels=256, out_channels=128)
```

to obtain a 128-dimension feature, which is then concatenated with all of the intermediate features. We finally decode the 1472-dimension extracted feature with

```
SharedLinear(in_channels=1472, out_channels=512)
SharedLinear(in_channels=512, out_channels=256)
SharedLinear(in_channels=256, out_channels=13, activation=False)
```

### A.1.3 PVCNN++ for Indoor Scene Segmentation

We also construct PVCNN++ based on PointNet++ [32] to demonstrate the general effectiveness of our PVConv. Concretely, it takes in 8192 points as input at each time and encodes the points with

```
PVConv(in_channels=9, out_channels=32, kernel_size=3, voxel_size=32)
PVConv(in_channels=32, out_channels=32, kernel_size=3, voxel_size=32)
SetAbstract(num_channels=[32, 32, 64], num_points=1024, radius=0.1)
PVConv(in_channels=64, out_channels=64, kernel_size=3, voxel_size=16)
PVConv(in_channels=64, out_channels=64, kernel_size=3, voxel_size=16)
PVConv(in_channels=64, out_channels=64, kernel_size=3, voxel_size=16)
SetAbstract(num_channels=[64, 64, 128], num_points=256, radius=0.2)
PVConv(in_channels=128, out_channels=128, kernel_size=3, voxel_size=8)
PVConv(in_channels=128, out_channels=128, kernel_size=3, voxel_size=8)
PVConv(in_channels=128, out_channels=128, kernel_size=3, voxel_size=8)
SetAbstract(num_channels=[128, 128, 256], num_points=64, radius=0.4)
SetAbstract(num_channels=[256, 256, 512], num_points=16, radius=0.8)
```

Then, the decoder upsamples the point cloud from 16 points to 8192 points using

```
FeaturePropagate(num_channels=[256, 256], num_points=64)
PVConv(in_channels=256, out_channels=256, kernel_size=3, voxel_size=8)
FeaturePropagate(num_channels=[256, 256], num_points=256)
PVConv(in_channels=256, out_channels=256, kernel_size=3, voxel_size=8)
FeaturePropagate(num_channels=[256, 128], num_points=1024)
PVConv(in_channels=128, out_channels=128, kernel_size=3, voxel_size=16)
PVConv(in_channels=128, out_channels=128, kernel_size=3, voxel_size=16)
FeaturePropagate(num_channels=[128, 128, 128, 64], num_points=8192)
PVConv(in_channels=64, out_channels=64, kernel_size=3, voxel_size=32)
SharedLinear(in_channels=64, out_channels=128)
SharedLinear(in_channels=128, out_channels=13, activation=False)
```

We refer the readers to Qi *et al*. [32] for the detailed definition of the set abstraction (SA) and feature propagation (FP) module. For each feature propagation (FP) module in the decoder, we concatenate the features from the corresponding set abstraction (SA) module with the same number of points.

### A.1.4  F-PVCNN for 3D Object Detection

For 3D object detection on KITTI [5], we construct our F-PVCNN based on F-PointNet [29], which is composed of an instance segmentation network, a center regression network and a box estimation network. We keep the center regression network and box estimation network the same as in Qi *et al*. [29], and we replace the primitives in the instance segmentation network with PVConv.

It first encodes the features using

```
PVConv(in_channels=3, out_channels=64, kernel_size=3, voxel_size=16)
PVConv(in_channels=64, out_channels=64, kernel_size=3, voxel_size=16)
PVConv(in_channels=64, out_channels=64, kernel_size=3, voxel_size=12)
PVConv(in_channels=128, out_channels=128, kernel_size=3, voxel_size=12)
SharedLinear(in_channels=128, out_channels=1024)
```

It then concatenates the final 1024-dimension feature, the max-pooled 1024-dimension feature over all points and the 3-dimension one-hot class label (*i.e*., car, pedestrian or cyclist) together. Finally, it passes the 2051-dimension encoded feature through the decoder:

```
SharedLinear(in_channels=2051, out_channels=512)
SharedLinear(in_channels=512, out_channels=256)
SharedLinear(in_channels=256, out_channels=128)
SharedLinear(in_channels=128, out_channels=128, activation=False)
```

An additional fully-connected layer processes the encoded features and makes a binary decision on whether a point belongs to the foreground or not.