[Reviews · NeurIPS 2019]

Reviewer 1



# Strong points * The paper is very well written and the problem clearly described. The illustrations are helpful. * The method is simple to implement, yet innovative and effective. * The experimental results support the proposed method, showing that this architecture achieves very good results with low memory overhead and low latency # Weak points * The experimental results contain no error bars nor a test for statistical significance. Many improvements over previous results are numerically quite small and it is unclear if these improvements are statistically significant. * The experimental evaluation sometimes appears more like an advertisement for the proposed method than an objective scientific evaluation of the pro and cons of the method. This includes the following points: - No baseline result (e.g. memory) was highlighted in the tables when best. - The grouping in Tab.1 and Tab. 3 appears a bit arbitrary and was apparently done in a way that the proposed method performs best compared to the baselines in each group. - Expressions like "outperforms [...] by a large margin" (ll. 258-259) appear more like advertisement than an objective description - Fig. 4 only compares latency against point cloud based method and memory against voxel based methods which appears unfair. Why not show voxel and point cloud based methods in both graphs? * Apart from Tab.2, the paper does not contain a proper ablation study (e.g. bilinear interpolation vs. nearest neighbor, higher resolution voxel grids in Tab. 2) * I am not a hundred percent confident about the comparison of latency and GPU memory in Tab. 1, 3, 4 and Fig. 4, as the baselines were not necessarily designed / tuned with these metrics in mind. It is hence not clear how much of the improvement is due to the concrete implementation vs. the proposed method. * It would have been nice to have a limitation section that discusses possible future work. # Questions / additional comments * Missing references (possibly also as baselines): [1,2] * How does the proposed method compare to sparse convolutions [3]? Maybe it would even make sense to combine the proposed method with sparse convolutions? * It was not clear to me why the memory of the proposed method grows sublinearly with the resolution of the voxel grid (ll. 241-242). Shouldn't the memory requirement also grow cubically (at least asymptotically), since the method in the end also uses 3D convolutions? * Sec. 5.3 only talks about a train and validation set. Is there no test set? # Overall rating This paper proposes a simple and effective new type of architecture for 3D deep learning. While the paper could still be improved at some place, I believe it is generally well done and should therefore be accepted. # References [1] Groh, Fabian, Patrick Wieschollek, and Hendrik PA Lensch. "Flex-Convolution." Asian Conference on Computer Vision. Springer, Cham, 2018. [2] Thomas, Hugues, et al. "KPConv: Flexible and Deformable Convolution for Point Clouds." arXiv preprint arXiv:1904.08889 (2019). [3] Graham, Benjamin, Martin Engelcke, and Laurens van der Maaten. "3d semantic segmentation with submanifold sparse convolutional networks." Proceedings of the IEEE Conference on Computer Vision and Pattern Recognition. 2018. === UPDATE AFTER REBUTTAL === I appreciate the authors' response (ablation study, comparison to sparse convolutions, updated Fig. 4) which answered most of my questions. I was only a bit disappointed that the authors still have not included a test for statistical significance in the rebuttal which would have been helpful given that the numbers are all relatively close. I am also a little bit confused by the statement that "the model has been evaluated 20 times to reduce the variance". Shouldn't every evaluation on the same test set (w/o retraining) give exactly the same result? It would be good if the authors could make this point more clear. All in all, I still believe that this is a good submission that should be accepted.

Reviewer 2



Pros: * Paper is well-written and easy to understand * Analysis of memory inefficiency of existing work is very thorough * Proposed solution is technically sound. * Solid experimental setting * Very strong results on multiple tasks Cons: * There is a lack of comparison study over state-of-the-art 3D detectors (voxel based) on KITTI test benchmark. * Other memory efficient convolution operators for sparse data, such as sparse-conv and SBNet should also be discussed / compared against [A, B]. [A] Graham B, Engelcke M, van der Maaten L. 3d semantic segmentation with submanifold sparse convolutional networks Proceedings of the IEEE Conference on Computer Vision and Pattern Recognition. 2018. [B] Ren, Mengye, et al. "Sbnet: Sparse blocks network for fast inference." Proceedings of the IEEE Conference on Computer Vision and Pattern Recognition. 2018. It would be great if the could also discuss how to extend other local operators under this two-stream setting, such as how we conduct local pooling, deconvolution, dilated convs, etc.

Reviewer 3



The paper presents impressive experimental results. The promises of high efficiency and good performance of the proposed method are certainly kept. For me, this combination of efficiency and good empirical performance is a main benefit of the proposed approach, which could make the proposed method interesting for a wider audience. Having said that, I still have some criticism about the paper. Regarding the experimental evaluation, I am missing ablation studies that would further investigate the effects of different design choices. This includes at least the following: - the effect of normalization vs. no normalization (L146-154) - the effect of the scale of voxelization, i.e. the voxel size before convolutions (L155-160) - devoxelization: the relative performance of hard assignment vs. trilinear interpolation (L166-168) - feature fusion: the detailed effect of residual addition vs. concatenation - the number of convolutions -- do we need the same number of voxel convolutions as previously point convolutions to be maximally effective (since the receptive field size will differ)? The paper currently only presents one version of the proposed approach, but a better understanding of those parameters and design choices is needed in order to effectively use the method in practice. In particular, I am not convinced the point cloud normalization will always be beneficial, since the normalization invariably loses information about the metric scale of objects. (Imagine, for example, two scans from the KITTI dataset: one where the recording vehicle stands in front of a wall and only perceives a depth range of 2-5m vs. a standard view along a street with a depth range of 2-50m. When normalizing both point clouds to the range [-1,1] and voxelizing them with a fixed voxel resolution, the voxels will correspond to very different metric sizes). Also, I would be particularly interested in the relative performance of the different options for devoxelization. While I follow the argument that both voxelization and devoxelization are differentiable, the nearest neighbor assignment version of voxelization/devoxelization will exhibit discontinuities if points are close to voxel boundaries. This could potentially lead to bad gradient flow and negatively affect network training. Regarding the indoor semantic segmentation experiment, the paper only reports results on S3DIS. In the 3D SemSeg community, S3DIS is considered outdated (and problematic because of differing evaluation protocols between papers). Instead, it is better to use ScanNet2, which allows for better controlled evaluation. Please comment on the points above in the rebuttal. Finally, I have some criticism about the writing and paper presentation - I found the writing style of the beginning of the paper a bit repetitive. By the time I had reached the end of Sec.3, I had the feeling that I had read the same sentences 3 times over. - The analysis of voxel methods only considers regular voxel grids. What about octree representations? They are far less memory intensive, and they should result in a more regular memory access pattern. - The experimental evaluation only considers baseline methods with regular voxel convolutions. [7] have shown that submanifold sparse convolutions are extremely effective -- with a variant of [7] currently leading on the ScanNet2 SemSeg benchmark. Yet, sparse convolutions are neither discussed nor compared against. Update It seems like we're all in line with our assessment of the paper. In my view, the rebuttal does a good job of clearing up the remaining points. I especially appreciate the detailed ablation studies. I will increase my score to Accept as a consequence.

[Author Response · NeurIPS 2019]

We thank all reviewers for their comments. All responses and changes will be incorporated into the revision. Code will be released for full reproducibility.

**Baselines (R1, R3, R4).** We compared PVCNN with SSCN (sparse-conv) and O-CNN (octree) as suggested by three reviewers. PVCNN is **4.8×** faster than SSCN with superior performance (Table 1). For O-CNN, the inference is fast; however, it is more than $10\times$ slower to preprocess the input data (*i.e.*, condense the point cloud and construct the octree) and postprocess the output (*i.e.*, refine the boundaries using the dense CRF). Also, as pointed out by R1, these two approaches are orthogonal to our method and can be combined together: replacing the volumetric convolution with the sparse/octree-based convolution.

|  | PVCNN | SSCN | O-CNN |
|---|---|---|---|
| Mean IoU | **86.2** | 86.0 | 85.9 |
| Preprocess | 0 ms | 0 ms | 86.9 ms |
| Inference | 50.7 ms | 241.8 ms | 5.3 ms |
| Postprocess | 0 ms | 0 ms | 842.4 ms |

Table 1: Results on ShapeNet Part.

**Scale Normalization (R4).** We *only* normalize the point cloud in the voxel branch while keeping the coordinates in the point branch unchanged; therefore, there is no information loss after two branches are merged together. Without instance scale normalization, the voxel grids are more than $10\times$ sparser on average, and the volumetric convolution is no longer effective to extract features. An alternative is to drop outlier points that do not lie in any voxel grids, which will inevitably induce information loss (see Table 2). Thus, the instance scale normalization is critical in the voxel branch.

|  | mIoU |
|---|---|
| PVCNN (on S3DIS) | **54.33** |
| w/o scale normalization | 52.62 |

|  | mIoU |
|---|---|
| PVCNN (on ShapeNet) | **86.2** |
| devoxelization w/o TI | 85.7 |
| feature fusion by concat | 86.1 |
| voxel resolution (0.75×) | 85.7 |
| voxel resolution (1.25×) | 86.1 |
| 1 voxel-conv per block | 85.6 |
| 3 voxel-convs per block | 86.1 |

Table 2: Ablation studies.

**Devoxelization (R1, R4).** We compared different implementations of devoxelization. From Table 2, the trilinear interpolation (w/ TI) performs better than the nearest neighbor (w/o TI), which is because the points near the voxel boundaries will introduce larger fluctuations to the gradient, making it harder to optimize, as mentioned by R4.

**Ablation Studies (R4).** More analyses are provided in Table 2, including different feature fusion methods, voxel resolutions (also see Table 2 in the paper), and number of convolutions per block. Our design choice is the best. We'll add them to the paper.

**Evaluation on S3DIS (R4).** We follow exactly the same data processing and evaluation protocol as PointCNN (L252) to make sure that the improvements are entirely from our proposed PVConv rather than different evaluation protocols. We thank R4 for suggesting the ScanNet2 dataset which we will experiment on.

**Evaluation on KITTI (R1, R3).** We choose F-PointNet as our baseline, which is a popular open-source 3D detection model. We do not compare with other state-of-the-art (voxel/point-based) models because the *region proposal network* has a large impact on the final results; we want to remove the influence of different region proposal networks and only evaluate the effectiveness of our *convolution primitive*. Besides, we do not present results on the testing set because F-PointNet only provides the 2D region proposals on the training and validation set.

**Feature Visualization (R3).** We illustrate the voxel and point branch features from the final PVConv in Figure 1. Note that warmer color represents larger magnitude. It is interesting to see that the voxel branch captures large, continuous parts while the point branch captures isolated, discontinuous details (*e.g.*, table legs, lamp necks). The two branches provide complementary information and can be explained by the fact that the convolution operation extracts features with continuity and locality.

(a) voxel branch    (b) point branch

Figure 1: Feature visualization.

**Statistical Significance (R1).** On KITTI, our model has already been evaluated for **20 times** to reduce the variance (L266). On ShapeNet and S3DIS, our results have relatively small variances: PVCNN achieves $(86.13 \pm \mathbf{0.04})\%$ in mIoU on ShapeNet, and PVCNN++ achieves $(58.95 \pm \mathbf{0.08})\%$ in mIoU on S3DIS (trained for **4 times** on both datasets). We want to emphasize that our goal is to achieve high accuracy *as well as* better efficiency. Some improvements over PointCNN might be small in accuracy (0.1% in Table 1); however, the speedup is significant (**2.7×**).

**Improvement Breakdown (R1).** The speedup comes from our better algorithm with lower complexity rather than the implementation: PVCNN can *theoretically* save the number of incontiguous memory accesses by $k$ times (where $k$ is the number of neighbors) and achieve better locality (L190-194). Both baselines and our models are implemented using well-optimized deep learning libraries (PyTorch and TensorFlow). We do nothing further to optimize the speed.

**Sublinear Memory Growth (R1).** Asymptotically, the memory indeed grows cubically; however, the voxel resolution of PVCNN is kept very low thanks to the high-resolution point branch (L246-247). With low resolutions (L242-244), the memory consumption is dominated by the point branch, not the voxel branch. Thus, the memory grows sub-linearly.

**Paper Presentation (R1, R4).** We will revise our paper to make it concise and objective. We would like to clarify that the groups are assigned by the accuracy in Table 1 and 3 in the paper, and we put our PVCNN's into groups to highlight the speedup with similar accuracy. As suggested, we also compared *both* point-based and voxel-based models (see right) to make Figure 4 in the paper more comprehensive.



[Meta-Review · NeurIPS 2019]

This work proposes an efficient method for processing 3D data in deep neural networks. The method is evaluated on competitive benchmarks and shows consistent improvements in efficiency while retaining or even improving predictive accuracy. The authors promise to make the code available. Three expert reviewers initially assessed the work as 7/8/6, with minor concerns. The authors provided a detailed rebuttal that was read and discussed by all reviewers. The final assessment is 7/8/7. This work makes a practically useful contribution to applying deep learning on 3D data and the analysis how the speedup is obtained is interesting to a larger audience.